# "*They need to feel at home not in a home.*" Experiences of residents and carers in mental health residential facilities: In-depth case studies from the Sedibeng district, South Africa

Samantha Mhlanga[1] [iD], Frances Griffiths[1,2], Lesley Robertson[3,4] and Jane Goudge[1]

[1]Centre for Health Policy, University of the Witwatersrand Johannesburg, Faculty of Health Sciences, Johannesburg, South Africa; [2]Medicine in Society, Warwick Medical School, Coventry, UK; [3]Department of Psychiatry, University of the Witwatersrand Johannesburg, Johannesburg, South Africa and [4]Community Psychiatry, Sedibeng District Health Services, Vanderbijlpark, South Africa

## Research Article

**Keywords:**
deinstitutionalisation; serious mental illness; community-based care; residential homes

**Corresponding author:**
Samantha Mhlanga;
Emails: samantha.mhlanga@wits.ac.za;
mhlanga54m@gmail.com

## Abstract

**Background:** Since the 1990s, the relocation of psychiatric patients from long-stay institutions to community-based supported living has increased globally. However, most evidence on suitable residential services comes from high-income countries, with little from low- and middle-income contexts. This study explored the experiences of residents and carers in three residential care homes for people living with serious mental illness in Sedibeng District, South Africa.

**Methods:** Three organisations were purposefully selected as in-depth case studies. Ninety-one face-to-face qualitative interviews were conducted with service providers, residents, and family members between October 2022 and June 2023.

**Findings:** Residents described severe psychosocial disability when living with families, but improved functioning in residential homes. Organisations 1 and 3 operated small 3–4-roomed houses in township areas, accommodating 21 and 40 residents respectively, who had community access and social interaction. In Organisation 3, residents formed romantic relationships, undertook paid work, and lived semi-independently. Organisation 2, a repurposed school-like building with four large dormitories for 86 residents, imposed strict movement controls; medication was used to manage behaviour, and caregivers reported safety concerns.

**Conclusions:** Smaller residential homes offer more autonomy and integration than large dormitory-style facilities. Policies and funding should support smaller, community-based supported accommodation for people with serious mental illness.

## Impact statement

The World Health Organization has called for the deinstitutionalisation of people living with serious mental illness (PLSMI), which means shifting from psychiatric hospitals to community-based care. However, in low- and middle-income countries (LMICs), progress has been slow due to a shortage of community-based supported accommodation services. There is limited evidence in LMICs on community-based supported accommodation services. In South Africa, most of these services are provided through residential care facilities. The residential care facilities are predominantly run by non-governmental organisations (NGOs) and staffed by lay people. Limited funding and resources may hinder NGOs from delivering high-quality care. In this study, we explored the experiences of residents living in the NGO-managed residential facilities and of the carers in providing care. We found evidence that the use of family-size units as residential care facilities provides opportunities to improve community-based services for PLSMI. Re-purposing school-like buildings to residential care facilities may lead to care that is similar to institutionalisation. Our study highlights the need to strengthen policies that support the implementation of family-size residential facilities as part of community-based mental health services.

## Introduction

Globally, there is an ongoing process of deinstitutionalisation, where psychiatric patients are shifted from hospital wards to receive community-based care (World Health Organization, 2022), which is supported by evidence suggesting improvement in social functioning, independent living skills, re-integration with the community and overall mental health outcomes (Harrison et al., 2020). The World Health Organization (WHO) advocates for recovery-oriented services (World Health Organization, 2021). Leamy et al. have developed a conceptual framework

used to support personal recovery of psychiatric patients (Leamy et al., 2023). The authors state that personal recovery involves five processes of <u>C</u>onnectedness, <u>H</u>ope and optimism about the future, <u>I</u>dentity, <u>M</u>eaning and purpose in life and <u>E</u>mpowerment (CHIME) (Leamy et al., 2023).

Most high-income countries (HICs) have made progress with deinstitutionalisation, where there are a range of accommodation services: residential care, supported housing and floating outreach services (McPherson et al., 2018). McPherson et al. developed the simple taxonomy for types of supported accommodation, which is defined by four domains, including staffing location, level of support, emphasis on move on and physical setting (McPherson et al., 2018). For example, type 1 accommodation has on-site staff providing a high level of support in communal settings with limited emphasis to move on (McPherson et al., 2018).

Deinstitutionalisation is progressing slowly in low- and middle-income countries (LMICs). Mental health policies and legislation in many countries need updating, and there is a shortage of community-based supported accommodation, funding, training and evidence-based interventions (World Health Organization, 2022). Non-governmental organisations (NGOs) often provide accommodation services in LMICs, providing access to mental health care, treatment and rehabilitation, enabling psychiatric patients to live a life of dignity, promoting community living and helping them be in the least restrictive environments (Thara and Patel, 2010; Visalakshi et al., 2023). However, several problems may limit NGOs from delivering a high standard of care, such as a shortage of adequate funding and trained mental health staff (Visalakshi et al., 2023). In addition, exploitation and abuse of psychiatric patients may occur in NGO-managed residential homes (Express News Service, 2018). In this article, we identified three NGOs in Sedibeng district, South Africa (SA), where we explored the experiences of residents living in the NGOs and of the NGO carers in providing care.

## Background

The South African National Mental Health Policy Framework and Strategic Plan 2023–2030 emphasises the need for mental health residential facilities (South Africa, 2023). The policy states that a recovery model should underpin residential facility services, including compliance with safety and human rights standards (South Africa, 2023). Residential facilities in SA are run by NGOs. The NGOs are predominantly run by lay people with lay caregivers responsible for providing daily care (South Africa, 2003). A 2009 modelling study by Lund and Flisher suggested that SA needs 107 beds in adult community facilities per 100,000 population (Lund and Flisher, 2009), and yet community facilities are under-resourced, providing 3.6 beds per 100,000 population (World Health Organization, 2007; Hassem and Laher, 2022).

In 2015, in an aim to cut costs, the Gauteng provincial government announced the termination of the contract between the Department of Health (DoH) and mental health facilities operated by a long-term provider, Life Esidimeni (Durojaye and Agaba, 2018; Robertson et al., 2021). As a result, 1,442 psychiatric patients had to be discharged from Life Esidimeni hospitals to alternative care facilities (Robertson et al., 2018; Robertson and Makgoba, 2018). There were insufficient existing facilities, so the government offered money to support NGOs to establish new facilities (Durojaye and Agaba, 2018; Robertson et al., 2021). Some patients were transferred to NGOs that were later found to be unlicensed, lacked resources and were inexperienced in providing mental

health care, resulting in the death of 144 patients (Durojaye and Agaba, 2018).

In response to the Life Esidimeni tragedy, the DoH introduced the licencing guidelines and district-based NGO governance and compliance teams (NGCTs) (Durojaye and Agaba, 2018; Robertson et al., 2021). The teams' role is to ensure NGOs meet the licencing requirements, provide training and ensure residents have access to quality care (Robertson et al., 2021).

## Methods

### Study approach and design

The study was qualitative, using a multiple case study approach (Gustafsson, 2017), to explore the differences and similarities of experiences in three NGOs (Gustafsson, 2017). Data were collected using a retrospective review of residents' care files, observations, face-to-face and telephone individual interviews. This design allowed us to triangulate our findings and enhance the credibility of our research (Gustafsson, 2017).

### Study setting

Sedibeng district in the Southern part of Gauteng Province has a population of 1,039,908 (Sedibeng district municipality, 2022). Sedibeng district currently has 14 NGOs providing 331 residential beds for people living with serious mental illness (PLSMI).

### Sampling and recruitment

#### NGO case studies
The NGCT provided us with a list of licenced NGOs, their size in terms of number of residents and how long they had been licenced for. We purposefully sampled three NGOs for diversity based on the description of the NGO environment, number of residents catered for, availability of staff and licencing status.

#### Retrospective record review
We developed a data extraction standard form using Excel sheet. This form was piloted, and SM reviewed early collected data with co-authors to check for consistency, completeness and relevancy. We included all care files of residents who gave us consent to review their files and extract data.

#### Observations and fieldnotes
SM spent 9 h a week undertaking observation at each NGO. In discussion with the NGO manager, SM spent 3 days a week observing a range of activities undertaken by the residents at different times of each day.

#### Interviews
*Managers*: We interviewed the managers of each NGO.

*Staff*: We used stratified random sampling to select staff based on whether they work during the day and/or night, how long they have been working for the NGO and the type of service they provide.

*Residents*: We met with each resident with a caregiver, where SM introduced the study and assessed their capacity to provide consent (see below) using their preferred language. If consent was obtained, we reviewed the case file and used stratified sampling to select residents based on gender and how long they had been staying at the NGO.

*Family members*: We used convenience sampling with managers from NGOs 1 and 3 sending a message on the family WhatsApp group. In Organisation 2, the manager asked the social workers to contact family members. Family members who were willing to participate gave the manager permission to share their contact numbers with us. None from NGO 2 agreed to participate.

Observations and recruitment for participants continued until data saturation was reached (Alam, 2021).

### Data collection

We developed a semi-structured interview guide for each type of study participant. The interview guides were piloted at an NGO, not included in the study and revised as necessary. All interviews were face-to-face, except for family members who were interviewed by telephone. Interviews were audio-recorded and conducted in English, Sesotho or IsiZulu languages. All participants gave consent to be recorded during the interviews. SM collected data from October 2022 to June 2023. We conducted participant observation (Lofland et al., 2022), participating in daily activities like serving food with the residents and caregivers, doing craft work and attending external social activities. Detailed notes were written every day.

### Data analysis

SM transcribed interviews verbatim. Interviews conducted using African languages were translated into English. SM double checked all the transcripts for accuracy. Transcripts and fieldnotes were read by all co-authors. We used NVivo 14 and MS Word to code the data. We used all data sources to describe the NGOs and the characteristics of residents. We used thematic analysis (Lochmiller, 2021) deriving our *a priori* codes from the CHIME framework of personal recovery (Trivedi and Mehta, 2019) and added emerging codes. We grouped coded data by NGO and participant type (managers, staff members, residents and families), and then compared data across NGOs and participant types.

### Ethics

The University of Witwatersrand Medical Ethics Committee granted us the ethics clearance certificate [M220402] and the Sedibeng district health approved the study. Having explained the importance of informed consent, all participants gave verbal and written consent to participate in the study, including to be recorded. We assessed residents' capacity to provide consent by explaining in a language of their choice the process of providing consent. We asked them to use their own words to reflect on what they had heard to demonstrate understanding. We only included residents who showed understanding of what was being asked of them, what consent means and that they were making their own independent decision to participate. All the transcripts were de-identified, replaced with codes and kept in a password-protected laptop.

### Results

#### NGOs

In Table 1 and Box 1, we summarise the characteristics of the NGOs. The three NGOs were established before the Life Esidimeni tragedy. The largest (Organisation 2) facility is exclusively for males. It is located in a school-like building with dormitories that can accommodate up to 21 people, and includes a large communal dining area. In the other NGOs, residents live in family-sized groups. In the smallest facility (Organisation 1), the residents of each group gather in a common room and eat together. In the medium-size NGO (NGO 3), residents have a choice to eat in their small group or in a communal dining area.

#### NGO residents

A total of 116 residents were assessed across all three NGOs. The ages of the residents ranged from 20 to 79 years, of whom 44 were Black, 3 were Coloured and 13 were White (Table 2). Most residents were diagnosed with schizophrenia. Some residents have arrived in the home after an admission from the hospital, and some were from their homes. Facility managers assess the residents to determine their eligibility for admission into their facility. The NGCT occupational therapist classifies the residents as high, medium and low functioning based on their ability to manage activities of daily living under minimum supervision and their cognitive awareness and orientation about themselves and their surroundings.

#### Residents, families and staff

Ninety-one interviews were conducted, of which 36 were residents (R), 17 family members (FM), 38 staff (3 managers [M], 3 administrators [A], 22 caregivers [CG], 3 cooks [CK], 2 nurses [N] and 5 cleaners [C]).

#### Experience before moving to an NGO

The feeling of hopelessness and loss of purpose in life prevents recovery. One resident described how he experienced suicidal thoughts before placement: "*I would drink paraffin and rattex. My brother would find me lying on the floor and take me to the hospital*" (NGO-2, R12). Another resident described how he became homeless: "*My mother passed away, and my stepfather chased me away. I became a street kid, picking food from the street because I was hungry. I will walk to go pick food from dirty places and come back in the afternoon*" (NGO-1, R5).

When PLSMI experience a lack of meaning in life and feel less empowered, they may become frustrated. This could result in them committing crimes: "*Eish! this is emotional. I was not aware that my brother takes weed. I don't know whether he was combining it with drugs. He was staying with my mother. So, my brother at one night raped my mother (crying). He got arrested and then came out of jail*" (NGO-1, FM4).

PLSMI that are struggling with recovery find it hard to participate in activities of daily living: "*His mother passed away, there was no one to take over. He was wild at home, didn't want to bath, change clothes. When I took him to the clinic, I had to go to the police station first, and call people from next door, and police will force him with guns to get inside the car*" (NGO-1, FM5).

#### Provision of basic needs

##### Medication

The manager from NGO 1 reported a case in which the medication was about to run out: "*The patient is taking lithium 1000. You cannot stay with that patient without medication. So, I said [to the nurses] they will rather admit her [to the hospital]. I only gave them two days. Eventually they brought the medication*" (NGO-1, M).

**Table 1.** Description of three NGOs providing accommodation services in Sedibeng district

| Services | | Description of services | NGO1 | NGO2 | NGO3 |
|---|---|---|---|---|---|
| Environment and resources | | Number of residents (number licenced for) | 21 (21) | 86 (72) | (40) (40) |
| | | Number of sleeping rooms/dormitories | 4 (3) | 4 | 9 |
| | | Number of users per sleeping room | 2–3 | 18–21 | 1–17 |
| | | Sickbay and laundry room | N | Y | Y |
| | | Activity room | N | Y | Y |
| | | Remote control gate, CCTV monitor | N | Y | Y |
| | | Running and back-up water and electricity | Y | Y | Y |
| | | Garden | Y | Y | Y |
| Services and activities | | On-site caregivers during the day | Y | Y | Y |
| | | On-site caregivers during the night | Y | Y | N |
| | | 3–5 meals a day, medication, clothes | Y | Y | Y |
| | | Recreational activities[a] | Y | Y | Y |
| STAFF | Professional | Manager/acting manager | 1 | 2 | 2 |
| | | Nurse/assistant nurse | 0 | 2 | 1 |
| | | Part-time social worker | 0 | 1 | 0 |
| | | Administrator | 1 | 1 | 1 |
| | Carers | Caregivers | 8 | 16 | 6 |
| | | Volunteers | 0 | 1 | 3 |
| | Housekeeping | Cook | 0[b] | 2 | 2 |
| | | Laundry and cleaners | 0[b] | 2 | 7 |
| | | Maintenance | 0 | 0 | 2 |
| | | Driver and/or supervisor | 0 | 1 | 1 |
| | | Security | 0 | 1 | 3 |
| Governance and licencing | | Board members | Y | Y | Y |
| | | Licenced[c] | Y | Y | Y |
| | | Acceptability certificates[d] | Y | Y | N |
| | | Health certificates[e] | Y | Y | N |
| | | Compliance certificates[f] | N | Y | N |
| | | Occupancy/zoning certificates[g] | Y | Y | N |
| Financial support | | Government subsidy | Y | Y | Y |
| | | Residents' social grants | Y | Y | Y |
| | | Donations | Y | Y | Y |
| | | Family contributions | N | N | Y |

*Note:* YES (Y), NO (N).
[a]Handiwork and crafts, games, going to church, gym exercises and gardening.
[b]Caregivers are responsible for cooking, washing and cleaning with help from the residents.
[c]Granted in terms of the Mental Health Care Act, 2002 (Act No. 17 of 2002), and its general regulations.
[d]Issued by Environmental Health Services in terms of the Foodstuffs, Cosmetics and Disinfected Act 1972 (Republic of South Africa, 1972).
[e]Valid health certificates issued by Environmental Health Services in terms of the National Environmental Health Norms and Standards.
[f]Certifying that the building meets all building regulation requirements issued by the municipality.
[g]Valid zoning or re-zoning certificate issued by the local municipality to indicate that the area has been inspected and deemed safe for occupancy.

Residents from NGO 1 appreciated help with taking their medication, which promotes hope for recovery: "*At home they were not taking care of me, I was just drinking them myself. Here at the centre, the manager fetches pills for me from the doctor, she encourages me to take the treatment, but at home I was their burden, you see*" (NGO-1, R8). A resident from NGO 3 appreciated the support she gets from the staff to take her medication: "*they give me my medication here because at home, I would get confused which medication to drink and sometimes forget to drink it*" (NGO-3, R5).

The manager in NGO 3 reported the additional burden placed on them as caregivers when the clinic changes medication: "*My concern with that is you've got residents that have been on olanzapine for many years and now are being changed to Seroquel® [quetiapine]. When they change from their meds [because there is no*

Box 1. Summary of the NGOs providing accommodation services.

**NGO 1**, founded in 2005, comprises three different houses in a township and is licenced for 21 residents. Each house consists of three bedrooms, a kitchen, a bathroom and a living room. The living room is used during the day for meetings and administering medication. During the night, the living room is used as a sleeping room for caregivers. Each house has two to four residents sharing a sleeping room. Caregivers stay at the residence for 2–5 days at a time and are responsible for medication, cooking, security and ensuring residents undertake personal care and activities of daily living. The administrator and the manager are responsible for taking residents to clinical appointments, supervising caregivers and day-to-day operations of the NGO.

**NGO 2**, operating since 1996, is licenced for 72 residents, although at the time of the study, it had 86 residents. The founder of the NGO was ill, and it was being run by an acting manager. It is located in a commercial area. There is a standalone office, a sickbay and an ex-school building. The school-like building consists of an activity room, a laundry room, four dormitories accommodating 18–21 residents, a toilet with rows of urinals and shower cubicles. The school hall is divided into a kitchen area and an eating area. Two caregivers sleep in each dormitory. The caregivers are responsible for taking residents to their clinic appointment. There is an on-site nurse and an assistant nurse working at the facility. They have a part-time social worker who supervises the caregivers, cleaners, cooks and laundry attendant and provides occasional counselling to residents in need.

**NGO 3**, founded in 2001, is licenced for 40 residents. There are 10 standalone buildings: a house each for the founder and the current manager, a multi-purpose house with 16 male residents, a cottage with shared bedrooms for women, two mobile homes for single female residents with a shared toilet, three buildings for residents living on their own, an administration office, a kitchen, a dining room and a sickbay. A part-time nurse assists a senior caregiver in managing and administering medication. The senior caregiver takes residents to their clinic appointments. Two residents work on rotation as security guards during the day, and a professional security guard works at night.

**Table 2.** Characteristics of residents who gave consent for the retrospective record review

| Characteristics of those who gave consent | | NGO1 | NGO2 | NGO3 |
|---|---|---|---|---|
| Gender | Males | 15 | 25 | 13 |
| | Females | 3 | 0 | 4 |
| Age (years) | 20–39 | 1 | 2 | 4 |
| | 40–59 | 13 | 18 | 12 |
| | 60–79 | 4 | 5 | 1 |
| Race | Black | 18 | 22 | 4 |
| | Coloured | 0 | 2 | 1 |
| | White | 0 | 2 | 12 |
| Education | No education | 1 | 0 | 0 |
| | Primary education | 7 | 1 | 9 |
| | Secondary education | 3 | 4 | 1 |
| | Not stated | 6 | 20 | 5 |
| Reason for NGO placement[a] | Referral from clinic/hospital | 9 | 8 | 9 |
| | Family struggling to provide care | 7 | 21 | 12 |
| Primary[b] diagnosis | Schizophrenia | 14 | 19 | 8 |
| | Schizoaffective disorder | 1 | 4 | 2 |
| | Bipolar disorder | 3 | 6 | 7 |
| Comorbid[c] conditions | Mental health conditions[d] | 6 | 4 | 6 |
| | Neuropsychiatric conditions[e] | 4 | 6 | 8 |
| | Other medical conditions[f] | 8 | 5 | 6 |
| Number of times residents have been hospitalised since NGO admission | 1–2 times | 7 | 11 | 7 |
| | 3–4 times | 2 | 0 | 1 |
| | 5+ times | 0 | 5 | 4 |
| | Never | 1 | 0 | 0 |
| | Not stated | 8 | 9 | 5 |
| Length of time at NGO | 1–5 years | 5 | 10 | 9 |
| | 6–10 years | 3 | 8 | 2 |
| | 11–15 years | 7 | 7 | 2 |
| | 16–19 years | 3 | 0 | 4 |

[a]Double count of residents, some residents had more than one reason for NGO placement.
[b]Some residents had more than one primary diagnosis.
[c]Some residents had more than one comorbid condition.
[d]Post-traumatic stress disorder, substance use disorder and borderline personality disorder.
[e]Epilepsy, dementia, intellectual disability, previous head injury and neuroleptic induced parkinsonism.
[f]HIV, TB, cardiovascular disease (include heart murmur and heart gallop), diabetes mellitus, dyslipidaemia, hypothyroidism and thrombocytopaenia.

olanzapine in stock], instead of a six-month review with the doctor at the clinic, we have to go every two months. So that, again puts pressure on our capacity, our transport, driving, the care workers" (NGO-3, M).

High medication use was reported in NGO 2, which carers linked to the need to manage residents' challenging behaviours. While this raised concerns about possible impacts on functioning and recovery of residents, the nurse described how they rely on medication to control behaviour: "*Most times the injection and medication are for when they are too high [hyperactive], others it helps them with, you find that they are sexually active, then it must suppress that thing*" (NG02-N). This is how the nurse perceived the role of medication, although the actual prescribing was carried out by the doctors, where residents go for their monthly checkups. Similarly, the caregiver said: "*If you see this one is being psychotic and fighting with everyone, we call a nurse to give him pills or injection, if he continues, we call the ambulance and we escort them to the hospital*" (NGO2-CG1). This shows how the carers perceive medication as important for managing the physical and sexual behaviour of residents. The residents complained that medication caused them to feel hungry: "*I feel like I have appetite when I take my medication. And there is not a single day where we don't get our medication*" (NGO-2-R10). The staff recognised this as a problem: "*The pills make them hungry. At the office they will give me either 12 kg or 10 kg of mealie-meal to cook pap. It is not enough but there is nothing I can do*" (NGO-2, CK). The caregiver stated: "*Mostly they fight for food. They don't get full, they will finish and want to steal for others. I think it is the pills, they make them starve*" (NGO-2, CG6). The manager reported that sometimes residents refuse their

medication: "*Others refuse to take medication at some point. They will tell you that they are tired; they have been taking medication. That is when they start being aggressive till we call the ambulance to take them to the hospital*" (NGO-2, M).

### Food

Access to food provides a sense of stability and reduces anxiety about starving. The caregivers from NGO 1 recognised the manager's effort to provide food: "*With (the manager) to be honest, she always balances. They aren't days when she will make them hungry, she buys for time ahead. If the admin goes out to buy food for the weekend, she will buy more. There aren't times when they are in shortage of food*" (NGO-1, Caregiver 2). However, some residents complained about the type of food. "*The food here, no man. Pap most of the time, mealie-rice, beetroot, cabbage, potatoes, soya mince. Why are we eating like this because at home, I am not eating such kinds of food*" (NGO1- R2). Poor quality of food could affect residents' recovery because they turn to feel disempowered and voiceless about the type of food they want to eat.

A shortage of food could make residents feel neglected by the service providers. Although the manager from NGO 2 reported that they provide meals to the residents: "*We are providing four meals per day*" (NGO-2, M). Four caregivers were concerned about the shortage of food: "*It is sad because at times they can give them soft porridge without sugar, at times pap without anything maybe with soup or cabbage so that they have something to eat*" (NGO-2-CG8). The residents complained about the amount of food they eat: "*Food is not enough, it is so little, and they starve you the whole the day without eating. Like even that porridge in the morning it is not enough. I keep drinking water when I get hungry. Sometimes, I have to organise my own food like instant porridge. If we have parents that come visit, they give us money and we organise our own food*" (NGO-2- R2).

The manager from NGO 3 described the amount of food they provide to the residents: "*We serve breakfast at half past seven, it's teatime at 10, main lunch is at one then it's three o'clock snack and juice and five o'clock is the last supper*" (NGO-3-M). The caregivers complained about food donations that expired: "*They should stop receiving expired donations. Like this year January, they were eating things that expired last year May 2022. These are things that come from [supermarket chain] in a black container. These patients vomit after eating*" (NGO-3-CG1). However, the residents appreciated their food, which gave them hope for a better quality of life: "*In the morning its either they give us porridge, matabella or oats. Oats is my favourite. They provide us with coffee, very strong sweet white coffee. Lunch we eat lovely, nice ox liver, taste like steak. They make it soft you know. Supper we only have sandwich because we get a huge lunch with three or four veggies*" (NGO-3-R1).

### Safety and security

A safe environment creates hope, connectedness and empowerment, which increases motivation for recovery. Residents from NGO 1 felt safe in residential homes: "*I like that we are safe. There isn't anyone that will rape us. Women and other people that are mentally sick, they are being raped*" (NGO-1, R8). One caregiver described how their community members respond to an unsafe situation: "*I feel safe, we have an alarm here. If the alarm goes on, our neighbours come and ask what is going on. If we need help, they are available to assist*" (NGO-1-CG2).

NGO 2 resident felt safe compared to when he was still in prison: "*I feel safe because it is not like prison. In prison, if you are locked up and there is a fight. The next person who is going to open that door might find you dead. Caregivers were told to not sleep fully here. Thank God for touch screen phones that keep them busy. If something happens, they quickly go and help*" (NGO-2-R5). However, caregivers from NGO 2 described their fear of sexual behaviour

when sleeping in the same dormitory with the residents: "*In our dormitory, we are only 2 women. We always see them busy masturbating while looking at you. Some will be imagining themselves with you. You just have to reprimand them, and they will tell you it is painful. There was a lady working alone and they tried going closer to her while she was sleeping. She reported but nothing happed. We are not safe at all*" (NGO-2-CG7). Some caregivers feared residents' physical violence: "*There is no safety here. You see this resident that just left here. He tends to fight others. He can even wake up at night and strangle you. You must always be on the lookout*" (NGO-2-CG8).

NGO 3 has a security guard working at night: "*It is very safe. They've got a security guard coming to check up on us quite often. He comes to check if anyone is awake and doing anything. That is good because I have a terrible fear thinking someone is going to come and kill me*" (NGO-3, R3). However, the caregivers described incidents of physical violence from the residents: "*When they become violent, you know you don't have a chance. I was hurt by a patient. I was holding him while he was kicking me in my stomach. He wanted to hit me with the cylinder, the fire extinguisher*" (NGO-3-CG5).

### Connectedness

#### Between residents

Positive relationships foster a sense of belonging and emotional support: "*As residents we create a lot of friendship through music, making jokes, talking about books*" (NGO-2-R2). As a result, they treat each other like siblings as the caregivers shared how residents are united: "*They are more like brothers and sister. You should see them doing the activities together, they help each other and take care of each other*" (NGO-1, CG6). One resident from NGO 1 reported how well he relates with other residents, even though their illnesses are not the same: "*With the people that I am staying with, other patients we get along very well. But we are not sick the same way, others are too sick because they talk alone, and others are alright, we speak well, play games just to push the sun to set*" (NGO1, R1). However, in NGO 2, the circumstances led to some residents fighting, which affects their emotional well-being. The Caregiver stated: "*The residents, eish when they are on their own, they fight. So, we should always be with them*" (NGO-2, CG4).

The staff members from NGO 3 described how residents have a good relationship with each other and the formation of romantic relationships: "*Most of the time residents get along. We've have got three couples, 2 that are married and one that's engaged*" (NGO-3, A). The caregiver stated, "*We have married couples, but they don't stay together. They are in different dorms*" (NGO-3-CG3). A healthy relationship encourages PLSMI to focus on their strengths as one resident was grateful to learn from her fiancé: "*We are a happy big family. I am so grateful to God to be in a place like this. I learn a lot from my fiancé. We get to spend money together. She writes the budget, and we buy our things together*" (NGO-3-R6).

#### Between residents and their families

Social relations with family members influence identity and how PLSMI perceive themselves. All NGO managers described the challenges they experience with the families of residents. NGO 1 manager described how they are struggling to get hold of some families: "*We never have any family visit, we are trying to reach the family of those who we know their home addresses, but we still have the families rejecting us*" (NGO-1-M). Residents described how they

felt abandoned by their family members. For example, NGO 1 resident stated: "*I want to go home, my mother and father passed away. I am left with my brothers and sister, they all have cars, but they cannot come here and see me*" (NGO-1-R3). The resident stated: "*Sometimes, I feel like my family abandoned me, they don't want me, and I am on my own you see*" (NGO-2, R12). Some residents long to go back home and be with family. For example, one resident described his effort to run away from the residential home: "*I have been here for too long. I told my sister that I want to come back home. She keeps making excuses. The manager won't let me go because they need my sister's permission. I ran away 3 times going to my sister. The staff found me there by the highway place*" (NGO-3-R8). NGO 1 family member described how the family environment is not conducive for the resident: "*I cannot stay with him, because I own a tavern here at home and it is always full of people drinking. People are going to give him alcohol and he will end up drinking. It is not safe where I stay*" (NGO-1, FM5).

The manager from NGO 2 described the challenges they experienced with residents who visit home: "*We encourage families to take the resident in December, and we also follow up with the families because the routine is not the same. Here they eat on time. They take medication on time. They sleep on time. So, we expect that they do the same at home. If not, the residents' relapse, and we have to send the social worker to assess the situation at home*" (NGO-2-M). The family member shared some challenges they experience when the resident visits homes: "*We have decided that we will not bring him home because last year December we brought him home for Christmas, but it was absolutely a disaster. First, we live in a secure environment, but he managed to get out and got lost and we found him about 6 and half kilometres from home, all dirty. It really didn't work out well*" (NGO-3-FM2). NGO 3 manager described how some families contribute to taking care of the residents: "*So, like I said, I probably have 12 families that give us a small family fee, but it's minority, most of my residents don't have family*" (NGO-3-M).

### Meaning in life and empowerment

Freedom of movement helps PLSMI create a meaningful life for themselves by participating in community activities: "*They need to allow residents to have freedom and move around. They need entertainment, parties to cheer them up. They need to feel at home not in a home. Being mentally ill doesn't mean that you are dumb. I might have my episodes, but they still need to do things they were doing at home or community. They are here to be monitored, take their medication not to be prisoners*" (NGO-2-CG3). NGO 1 manager allowed high-functioning residents to go outside the residential facility: "*Most of my residents are high-functional, because we can send them to the shops*" (NGO-1-M). One resident from NGO 1 confirmed that he is allowed to go to the shops on his own: "*I am able to go out alone when they sent me to the shops*" (NGO-1-R9). The manager from NGO 3 described how some of the residents can go to the mall on their own: "*Those who can [go to the shopping mall], those who can't don't. We know our residents. For instance, this one resident is schizophrenic but capable. Once a month, he will say his sister sent him R400 and he wants to go to the mall for a pizza with a friend. So, I will say no problem, but they will send me a screen scrab of the uber driver photo and car registration. They will also let me know when they arrive or leave*" (NGO-3-M). The caregiver from NGO1 described how they must supervise residents who go out: "*So, at times they need snacks, they are patients that are allowed to go out…depending on their condition… These 2 can go to buy but*

*we monitor them. You must wait by the gate; they go buy and then come back…*" (NGO-1-CG3).

However, NGO 3 residents are only allowed to go out when they have permission: "*As a security [also a resident], I shouldn't allow any resident to go out unless they have the permission to go home or go to the mall with their family*" (NGO-3-R11). The caregiver explains why residents are no longer allowed to go out on their own: "*They were allowed to go but they stopped it because they used to go buy drugs, some will go to the mall without saying anything and the manager will find them there*" (NGO-3-CG6).

Restricting movement can hinder PLSMI's ability to participate in meaningful activities that promote recovery. The administrator from NGO 2 stated: "*We don't allow patients to go to the shops alone because these are people who are sick. You will think they are going to the shops while they are running away. When they run away, it becomes the manager's and the caregivers' responsibility. Their families know that they are here, when they get lost, we must account for that*" (NGO-2-A). As a result, the residents from NGO 2 felt like they were in prison: "*I don't want to be in this place anymore, we are always locked. They shouldn't keep people locked here like they are criminals, like we committed a crime. Even in prison they open for prisoners to go out and do some work. When I tell them I want to go, they say they want parents. I don't have parents*" (NGO-2-R-6). The other resident stated: "*The caregivers are guarding us always. They sleep with us in dormitories. They have been guarding me for 11 years. That is prison sentencing*" (NGO-2-R4).

Allowing PLSMI to make decisions about their life promotes empowerment. Some residents from NGO 3 had their own rooms: "*My mother and brother bought me some tiles and paint for my room. They bought me shelves and the TV as well. I just need to start painting my room and put tiles because I don't like carpet*" (NGO-3-R5). Some residents were empowered to work, which provides a sense of purpose in life: "*I asked for a job, and they gave me. I work at the gate. At 5:00 o'clock I wake up by myself, get ready for work. The night guard will open the dorm gate for me, we walk together to the gate, and I just sit and work till 1730 h*" (NGO3-R6).

### Discussion

In this study, we explored the users' and carers' experiences of three different NGO-managed residential care homes in the Sedibeng district of Gauteng province, South Africa. In sum, the living arrangements made a significant difference to the type of care provided and the experience of care. Two of the NGOs had family-size units, providing an experience similar to living in a normal home, while the third one had big dormitories in a school-type building. In the latter, there was high use of medication (increasing residents' appetites), which appeared to be used to control disruptive behaviour. Caregivers felt unsafe as they slept in the dormitories with the residents. Residents were not allowed to go outside the NGO. In contrast, residents in family-size units were satisfied with their residential services and the amount of freedom. Those who were classified as high-functioning residents by the occupational therapist were able to go to the nearest shops. Some residents had romantic relationships with other residents, engaged in paid work and lived on their own in a small house in the grounds of the organisation, which allowed them privacy.

Since 2006, the United Nation Committee on the Rights of Persons with Disability has emphasised the importance of deinstitutionalising people with mental illness to live in a home-like environment that will enable them to have freedom, live

independently and participate fully in society (United Nations, 2006; UN Committee on the Rights of persons with Disabilities, 2018). In our study, we found evidence that residents in family-sized units socialised with each other and had some degree of freedom of choice. They engaged with their local communities, and some had paid work. This is similar to the findings of a study conducted in India, where they moved 11 residents from a psychiatric institution to supported housing (Padmakar et al., 2020). Similar to residential facilities 1 and 3 in our study, the supported housing unit in India accommodated fewer residents, with staff on site to assist residents with daily living activities. After 6 months, residents were more sociable and they were selling vegetables and flowers to the local community (Padmakar et al., 2020). In Brazil, residential facilities were introduced that accommodated 10 people each and provided support for up to 6 months after long-term psychiatric hospitalisation (Marchionatti et al., 2023). By contrast, the residential facilities in our study did not have a time limit on the length of stay, with many residents having lived there for several years. Despite these differences, both models from Brazil and South Africa used smaller facilities that promoted community integration, autonomy and person-centred care (Marchionatti et al., 2023).

The National Mental Health Policy Framework and Strategic Plan 2023–2030 highlights the need to protect the human rights and dignity of people with mental illness (South Africa, 2023). It calls for mental health services that are recovery-oriented and respect the autonomy of the mental healthcare users (South Africa, 2023). However, South Africa still lacks residential facilities (South Africa, 2023), and this shortage has resulted in some residential facilities taking more residents than they can accommodate, and using unsuitable facilities. For example, in our study, the school-like building was catering for 86 male residents, and they relied on lay people with no professional training to provide care.

The severe behavioural disturbances reported in our study, which often led to the use of high doses of medication, were at least partially caused by under-resourced institution-like care provided. Although it may also be related to the inappropriate placement of people whose conditions were not stable in a supported living environment when they needed hospitalisation, it may be that the residents would have more control over their own behaviour if living in a smaller, more personal environment. Two studies from Australia found that in large licenced boarding facilities, residents had no control over their lives, they were socially isolated and experienced a shortage of resources (Drake, 2014; Drake and Herbert, 2015). These licenced boarding facilities were described as a process of transinstitutionalisation, where PLSMI are moved from psychiatric institutions to community care facilities that are similar to an institution (Drake, 2014; Drake and Herbert, 2015). Shortage of resources left residents feeling frustrated, often resulting in disruptive behaviour (Deane et al., 2012; Drake, 2014; Drake and Herbert, 2015). Similarly, in assisted living facilities with high numbers of residents in the United States, problematic behaviour affected the quality of everyday life and autonomy of residents (Morgan et al., 2016).

In England and Wales, the Mental Capacity Act of 2005 provides safeguards to protect people who lack the capacity to make decisions (Shah et al., 2011). This Act includes the deprivation of liberty safeguards, which ensures that every restriction placed on people with mental health are necessary and in their best interest (e.g., promote safety). The National Institute for Health and Care Excellence's (NICE) guidelines further emphasise that service providers of supported accommodation should understand how to assess mental capacity before making decisions on residents' behalf (National Institute for Health and Care Excellence [NICE] Guidelines, 2020). In cases where service providers should make decisions for people suffering from mental illness, they should be able to follow the necessary steps as highlighted in the Mental Capacity Act (National Institute for Health and Care Excellence [NICE] guidelines, 2020).

In South Africa, there is confusion in the National Policy for NGO licencing, which comments on people being admitted to NGO residential facilities under the Mental Health Care Act, but does not require any qualifications among the personnel (South Africa, 2002). In addition, the NGO residential facilities form part of the Community Mental Health Services as described in the National Mental Health Policy Framework and Strategic Plan. Community Mental Health Services care for people after discharge from the hospital if they cannot be cared for by primary health care (PHC) and their families (South Africa, 2002). These centres are intended to facilitate community living and integration for people who are stable. Therefore, these centres are not hospitals, and all the residents are outpatients at the community psychiatric and PHC clinics in the district, and so admission under the Mental Health Care Act cannot apply as the forms require justification of inpatient care. In theory, all the residents stay at the NGOs voluntarily. However, many might not have the capacity to make truly informed decisions and are at NGOs at the recommendation of a social worker or their family, possibly related to having no alternative accommodation and needing support in their activities of daily living. In addition, the large impersonal NGOs are more reminiscent of restrictive mental institutions (albeit with unqualified caregivers rather than psychiatric nurses) than supported living.

Our findings highlight challenges such as sexual harassment experienced by caregivers in mental health residential facilities that accommodate a large number of residents. Similar concerns were reported in a study from Denmark, which found that sexual harassment often occurred during everyday interactions with residents, leaving caregivers in vulnerable and unsafe positions (Nielsen et al., 2017). Many workers tend to normalise or downplay harassment, viewing it as "part of the job," which silenced their voices and limited reporting. At the same time, caregivers carry a heavy workload with limited training, supervision and job security, which places them at risk of stress and burnout (Nielsen et al., 2017). Harry and colleagues, in their systematic review and meta-analysis, also identified factors associated with caregiver stress and burnout, including low job satisfaction, feeling unsupported and staff shortages (Costello et al., 2019). Caregiver burden and risks to sexual safety highlight the need to strengthen policies that will protect caregivers providing care in these mental health residential settings.

Our findings show that some residents across all facilities were longing for social relations with their family members. Facility managers also described the difficulties they face when trying to engage with families in providing support. According to Ntsayagae et al. (2019), family caregivers often experience physical and emotional exhaustion while caring for a relative with a mental illness (Ntsayagae et al., 2019). Some families feel powerless due to limited knowledge about mental illness (Ntsayagae et al., 2019). These challenges may discourage families from wanting to live with or directly care for a relative with mental illness. As a result, some families seek alternative care arrangements, allowing their relative to receive support elsewhere while they distance themselves to avoid the emotional burden of caregiving.

## Strengths and limitations

Strengths of this study include the selection of three different NGO-managed residential care homes for PLSMI, and the exploration of the lived experiences of multiple participants, including residents, families and the service providers. Limitations of the study include the fact that the experiences and views of family members from Organisation 2, non-consenting residents, are still not known, and that the study was restricted to licenced NGOs in only one district of South Africa. The shortage of empirical studies from LMICs made it challenging to compare our findings and limited the generalisability of results.

## Implications for policy and research

The use of family-size units as residential care facilities in South Africa provides opportunities to improve community-based care services for PLSMI. Our study demonstrates this can be achieved within current licencing and resourcing in South Africa. Establishment of similar care facilities will help meet the World Health Organization's target to double community residential care facilities by 2030 (World Health Organization, 2022). In addition to strengthening residential-based care, there is a need for structured programmes to prepare and support families in caring for relatives with serious mental illness. Further research is needed in LMICs regarding the level of disability among residents, as well as the role and resourcing of residential care facilities. In South Africa, there is a need for research to understand the extent of unlicensed provision and barriers to gaining a licence.

## Conclusions

Our study provides evidence that organisations with fewer residents and smaller residential types of houses facilitate better living conditions than those with a high number of residents in large buildings. NGO 2 exemplifies resource maximisation in low-resource settings. However, re-purposing school-like buildings to residential homes may lead to tighter controls, overmedication and lack of contact with the surrounding community, leading to care that is similar to institutionalisation. Such re-purposing should be done to avoid compromising care, dignity and safety. There is a need to strengthen the implementation of family-sized residential units, and that will help enable PLSMI to exercise their human rights.

**Open peer review.** To view the open peer review materials for this article, please visit http://doi.org/10.1017/gmh.2025.10077.

## Abbreviations

| | |
|---|---|
| HICs | high-income countries |
| LMICs | low- and middle-income countries |
| NGO | non-governmental organisations |
| PLSMI | people living with serious mental illness |
| SMI | serious mental illness |
| WHO | World Health Organization |

**Data availability statement.** The dataset is available on a reasonable request. The request to access the dataset can be directed to Professor Jane Goudge, email: Jane.Goudge@gmail.com, and Samantha Mhlanga, email: mhlanga54m@gmail.com.

**Acknowledgements.** The authors would like to thank the following individuals and organisations for their invaluable contribution and support: Sedibeng NGC teams, NGO managers, staff, residents, family members, Sedibeng district health services, Wits, Centre for Health Policy and South African Research Chairs Initiative (SARChI).

**Author contribution.** All authors contributed to conceptualising the research project. SM collected, analysed and interpreted the data and wrote and revised the manuscript with JG, FG and LR. All the authors critically revised the manuscript and approved the final manuscript as submitted.

**Financial support.** The manuscript is part of a PhD project funded by the South African Research Chairs Initiative (SARChI) under the Department of Science and Technology (DST) and National Research Foundation (NRF), reference number MND210619613383.

**Competing interests.** The authors declare none.

**Ethics statement.** The University of Witwatersrand Medical Ethics Committee granted us the ethics clearance certificate [M220402] and the Sedibeng district health approved the study. All participants were provided with verbal and written consents. All the transcripts were de-identified, replaced with codes and kept in a password-protected laptop.

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
