## [Reviewer Report]

Strengths

• Grounded in Fieldwork

o 91 qualitative interviews across diverse stakeholders (residents, families, staff) provide rich experiential data.

o Triangulation through file reviews, interviews, and observations enhances reliability and depth.

• Human Rights & CHIME Lens

o Skillful application of the CHIME recovery model highlights core dimensions of autonomy, identity, and empowerment.

o The study reveals stark contrasts between institution-like care and family-sized, dignified environments.

• Policy Relevance & Urgency

o Clear implications for national mental health policy (e.g., South Africa’s NMHPF-SP 2023–2030).

o Situates findings within global deinstitutionalization trends and echoes WHO calls for rights-based, community-linked care.

• Comparative Case Study Design

• The juxtaposition of three NGO-run facilities illuminates how structural differences affect recovery outcomes.

• NGO 2’s dormitory-style housing serves as a cautionary tale, while NGOs 1 and 3 illustrate scalable, inclusive alternatives.

Gaps & Areas for Improvement

• Theoretical Anchoring & Cross-National Analysis

o While CHIME is used as a coding frame, the integration of other frameworks (e.g., structural violence, intersectionality) could deepen analysis.

o More systematic comparisons with LMIC contexts (e.g. India, Brazil, Kenya) would strengthen generalizability.

• Underexplored Themes

o Gender and Sexual Safety: Female caregivers’ experiences with harassment deserve deeper unpacking.

o Staff Capacity & Burnout: Lay caregiver burden is noted but not critically examined; links to supervision, mental health, and job security are limited.

o Family Dynamics: Residents' longing for familial connection merits exploration beyond logistical challenges — e.g. emotional labour, intergenerational trauma. Families, especially those on the lower end of the economic continuum also have significant challenges that impede care. Even simple things like how a person eats their meal in a shared home may differ to the cultural practices of a family- therefore families need to be trained and prepared to accept people back.

o The economics of these models and how they are resourced also have huge implications on the extent to which they will be scaled, these do not find much discussion in the paper.

• Ethics and Power

• Medication as behavioural control is flagged, yet ethical analysis is restrained. A stronger critique could question coercive practices and informed consent norms.

• The deprivation of liberty parallels incarceration — but the legal/rights-based implications are only briefly discussed.

• NGO 2 exemplifies how resource-maximizing strategies that may be important in low resource settings, however this repurposing needs to be done carefully without compromising care, dignity and safety.

• Global Mental Health Advocacy

This study aligns with UNCRPD mandates and WHO Mental Health Atlas priorities.

Resonates with advocacy for minimum service standards, family reunification support, and culturally sensitive care environments.

It strengthens calls for differentiated care models, embedded governance structures, and user-led monitoring frameworks.

There’s room to build on its insights to frame a South-South learning exchange — spotlighting India’s innovations (e.g., The Banyan and the work of Udaan under Tata Trusts in Nagpur) alongside South African models.

---

## [Reviewer Report]

A well written manuscript of a rigorous qualitative research titled: "They need to feel at home not in a home.” - Experiences of residents and carers in mental health residential facilities: In-depth case studies from the Sedibeng district, South Africa.

Method:

The study is premised on a well presented background/justification, and the authors carried out a scientifically rigorous qualitative exploratory case studies of the experiences of residents, carers and managers in 3 purposefully selected NGO operated mental health residential facilities in Sedibeng district, as they set out to do.

Results:

The findings justify the adequacy of the qualitative research procedure adopted. The methodological and data source triangulation provided credibility, validity, and depth of understanding on the research question.

The authors however failed to explore or discuss the adequacy of current funding for the operations of the NGOs, despite being one of the research questions as a possible limitation to the quality of services provided by NGOs, in addition to lack of trained mental health support staff. The only suggestive report was the remark by the manager of NGO 3 (line 319) - that only minority of residents' families support the NGO with small family fees.

Conclusion:

The authors concluding recommendations regarding funding and policy are not explicit. Are the observed service limitations attributable to challenges of funding and/or policy regulation and compliance?

Report on adequacy of current funding is needed for a holistic qualitative appraisal of NGOs policy compliance as well as the efficiency of their services. It would also enhance the utility of this study findings as evidence-based resource for mental health service improvement in LMICs